# Genistein Regulates Lipid Metabolism via Estrogen Receptor β and Its Downstream Signal Akt/mTOR in HepG2 Cells

**DOI:** 10.3390/nu13114015

**Published:** 2021-11-10

**Authors:** Hong Qin, Ziyu Song, Horia Shaukat, Wenya Zheng

**Affiliations:** Department of Nutrition Science and Food Hygiene, Xiangya School of Public Health, Central South University, 110 Xiangya Road, Changsha 410078, China; qinhong@csu.edu.cn (H.Q.); song0305ziyu@126.com (Z.S.); hooriyashaukat@gmail.com (H.S.)

**Keywords:** genistein, hepatic lipid metabolism, estrogen receptor β, Akt, mTOR

## Abstract

Genistein (GEN) has been shown to significantly inhibit hepatic triglyceride accretion triggered by estrogen deficiency. The main purpose of this in vitro study was to investigate the function and molecular mechanism of estrogen receptor β (ERβ) in regulating hepatic lipid metabolism induced by GEN. Different doses of GEN or GEN with an ERβ antagonist were treated with HepG2 cells. Results showed that 25 μM GEN significantly diminished triglyceride levels. Meanwhile, GEN downregulated the levels of genes and proteins involved in lipogenesis, such as sterol-regulatory element-binding protein-1c (SREBP-1c), fatty acid synthase (FASN), and stearoyl-coenzyme A desaturase 1 (SCD1), and upregulated the gene and protein levels of the regulation factors responsible for fatty acid β-oxidation, such as carnitine palmitoyltransferase 1α (CPT-1α) and peroxisome proliferator-activated receptor α (PPARα). Furthermore, 25 μM GEN reduced the levels of phosphorylation of protein kinase B (Akt) and mechanistic target of rapamycin (mTOR). Moreover, most of these effects from GEN were reverted by pretreatment with the antagonist of ERβ. In conclusion, GEN improved hepatic lipid metabolism by activating ERβ and further modulation of Akt/mTOR signals. The results provide novel aspects of the regulatory mechanism of ERβ on hepatic lipid metabolism and might help to profoundly understand the functions of food-derived phytoestrogens in preventing and treating hepatic steatosis in postmenopausal women.

## 1. Introduction

Postmenopausal women tend to have a higher risk of developing lipid disorders due to deficient internal estrogen levels. As the liver plays a key role in metabolizing lipids, hepatic steatosis indicates an enhanced risk of life-threatening ailments, including hyperlipemia, hypertension and atherosclerosis. The Dallas Heart Study, consisting of 1018 women aged 30–65, found that postmenopausal status has a direct impact on increased prevalence of hepatic steatosis, as well as an increased absolute hepatic triglyceride content [1]. Other research showing data from 488 postmenopausal women summarized that the duration of menopause directly impacts the severity of liver disease with increased risk of hepatic fibrosis or even hepatocarcinoma [2]. As estrogen directly influences energy homeostasis [3], hormone replacement therapy (HRT) has been utilized as a treatment for preventing postmenopausal women from developing lipid disorders [4]. However, it is still controversial whether HRT might be associated with an increased risk of breast and endometrial cancers [5,6]. In addition, currently, there is still no precise pharmacotherapy approved for hepatic steatosis; therefore, researchers are implementing knowledge on food nutrition to alleviate lipid disorders [7] and to further improve quality of life, which could be an effective strategy to block the progression of hepatic steatosis in postmenopausal women.

Genistein (GEN) is one of the most widely distributed dietary phytoestrogens and is found as a major class of compounds in soybeans. It has a structural resemblance to 17β-estradiol (E2) and exhibits weak estrogenic activity in mammals [8]. GEN has been widely discussed in the context of its role in disease prevention [8]. It is evident from published data that GEN exhibits beneficial effects in preventing metabolic diseases, more precisely postmenopausal metabolic syndrome, in both in vitro and in vivo model studies [9,10]. Estrogens play a role in terms of physiological effects, mainly via two estrogen receptor (ER) subtypes, ERα and ERβ. Several studies have pointed out that GEN shows preferential binding affinity to ERβ [11], whereas E2 regulates lipid metabolism mainly through ERα. A previous study by Weigt et al. [12] proved that ERβ is expressed in ovariectomized rat liver tissues, and the imposed inhibitory effects on the hepatic lipogenesis of GEN was found to be similar to that of ERβ agonist treatment, establishing that GEN might induce inhibitory impacts on triglyceride accumulation in the liver via ERβ under estrogen deficiency. However, whether these inhibitory effects are regulated by ERβ has not yet been fully validated, and the underlying molecular mechanism of ERβ in regulating hepatic lipid metabolism needs to be investigated further.

The mechanistic target of rapamycin (mTOR) signaling pathway regulates many metabolic and physiological processes in different organs or tissues, which emphasizes its role in modulating lipid metabolism in the liver [13]. mTOR is an evolutionarily conserved serine/threonine kinase that belongs to the phosphoinositide 3-kinase (PI3K)-related kinase family [14] and has been reported to play a critical role in promoting lipogenesis by regulating the expression of many lipogenic genes [15]. As one important family of transcription factors, mTOR controls sterol-regulatory element-binding proteins (SREBPs). Moreover, hyperactivation of mTORC1, one of the two distinct mTOR signaling complexes, induced expression of lipogenic and lipoprotein assembly genes (microsomal triglyceride transfer protein (MTTP) and apolipoprotein B (ApoB)), thereby elevating cellular triglycerides (TG) and diminishing secretion of ApoB-containing TG-rich lipoproteins [16]. There are studies demonstrating that activation of ERβ was involved in the prevention and treatment of cancer progression by modulating the PI3K–protein kinase B (Akt)/mTOR signaling pathway [17,18,19]; however, the functions of ERβ and the correlation between ERβ and Akt/mTOR in regulating hepatic lipid metabolism are not yet clear. Hence, this study is aimed at investigating whether GEN alleviates lipid disorders caused by estrogen deficiency in the liver through ERβ and further regulation of mTOR signaling.

## 2. Materials and Methods

### 2.1. Chemicals and Reagents

GEN and E2 were purchased from TargetMol (≥99.5%, Boston, MA, USA). PHTPP (99.64%, an ERβ antagonist) was purchased from Med Chem Express (Monmouth Junction, NJ, USA). Dulbecco’s modified Eagle medium (DMEM), penicillin/streptomycin solution, Trizol, RIPA lysis buffer, protease inhibitor, 1% phenylmethylsulfonyl fluoride (PMSF), a bicinchonininc acid (BCA) protein assay kit and a thiazolyl blue tetrazolium bromide (MTT) assay kit were purchased from Ding Guo Changsheng Biotechnology Co. Ltd. (Beijing, China). DMEM without phenol red was purchased from Procell Life Science & Technology Co., Ltd. (Wuhan, China). Fetal bovine serum (FBS) was purchased from Gibco of Thermo Fisher Scientific (Waltham, MA, USA). Certified charcoal/dextran-stripped foetal bovine serum (c/d-FBS) was purchased from Biological Industries (Kibbutz Beit Haemek, Israel). TG assay kits were obtained from Jiancheng Bioengineering Institute (Nanjing, China). Protein loading buffer was purchased from NCM Biotech (Suzhou, China). Goat anti-Rabbit IgG (H&L)-HRP and Goat anti-Mouse IgG (H&L)-HRP were purchased from Bioworld Technology (Bloomington, MN, USA). ERβ was obtained from Santa Cruz Biotechnology (Dallas, TX, USA). Phospho-mTOR-ser2448 (p-mTOR), AKT1, and Phospho-Akt-Ser473 (p-Akt) were purchased from ZEN-BIOSCIENCE (Chengdu, China). mTOR, fatty acid synthase (FASN), peroxisome proliferator-activated receptor α (PPARα), carnitine palmitoyltransferase 1 (CPT1α), MTTP, stearoyl-coenzyme A desaturase 1 (SCD1) and β-actin were purchased from Abclonal (Woburn, MA, USA). SREBP-1c was purchased from Proteintech (Wuhan, China).

### 2.2. Cell Culture and Treatment

HepG2 cells were obtained from the Peking Union Cell Center (Beijing, China). Cells were grown in DMEM supplemented with 10% FBS and 1% penicillin−streptomycin solution in a humidified incubator at 37 °C in 5% CO_2_. Cells were disassociated when the confluence of adherent cells reached 80–90%. To eliminate the environmental estrogenic effects before each experiment, cells were passaged at least once in DMEM without phenol red containing 10% c/d-FBS. Then, HepG2 cells were treated with dimethyl sulfoxide (DMSO), E2 (1 nM), or GEN (1, 10, 25 µM) for 24 h. In the trials with the ERβ antagonist, HepG2 cells were pretreated with 1 µM PHTPP for 2 h prior to GEN treatment.

### 2.3. Cell Viability Assay

Proliferation of HepG2 cells after E2 or GEN treatment was verified by MTT assay to study the effects of E2 or GEN on cell proliferation and to determine the intervention concentrations. HepG2 cells were seeded in a 96-well plate at a density of 5.0 × 10^3^ cells per well. After cell adherence, cells were treated with variable concentrations of GEN (0.01, 0.1, 1, 10, 50, 100 µM), E2 (0.1, 1, 10, 100 nM), or PHTPP (0.01, 0.1, 1, 10, 50 µM) for 24 h. After termination of the intervention, the old medium was discarded with 90 µL of new medium and 10 µL MTT in each well for 4 h. The medium was then removed and 150 μL of DMSO was added. The medium was shocked for 10 min to fully dissolve the crystallization; then, the absorbance was measured with a microplate reader at 490 nm.

### 2.4. Lipid Content Assay

Intracellular TG content was determined utilizing a TG assay kit and normalized to total intracellular protein using a BCA kit according to the manufacturer’s protocol.

### 2.5. Western Blot Analysis

HepG2 cells were harvested with RIPA lysis buffer encompassing a protease inhibitor and 1% PMSF. The protein samples (35 µg) were separated by 8 or 10% sodium dodecyl sulfate polyacrylamide gel electrophoresis (SDS−PAGE), and the gels were then run at a constant voltage of 80V and transferred to polyvinylidene fluoride (PVDF) membranes at a constant voltage of 80V. The membranes were immunoblotted overnight at 4 °C using primary antibodies specific to mTOR (1:1000), p-mTOR-Ser2448 (1:750), SREBP-1c (1:750), SCD1 (1:1000), MTTP (1:1000), PPARα (1:1000), AKT (1:1000), p-AKT (1:750), FASN (1:1000), ERβ (1:750), CPT1α (1:1000), and β-actin (1:300,000). Bands were incubated with secondary antibody for 1 h at room temperature. Immunoreactive bands were visualized via electrochemiluminescence (ECL) and quantified using a chemiluminescence imager (Tanon-5500, Shanghai, China). Bands were quantified via densitometry by Image J.

### 2.6. Quantitative Real-Time Reverse Transcription Polymerase Chain Reaction (RT-PCR)

Total RNA from HepG2 cells were extracted utilizing a Trizol reagent, and cDNA was synthesized using HiScriptIIQ RT SuperMix (Vazyme, Nanjing, China). Gene expression was quantified using Hieff UNICON qPCR SYBR Green Master Mix (Yeasen Biotech, Shanghai, China) and a LightCycler 480 II (Roche, Basel, Switzerland). Relative gene expressions were calculated using the 2-ΔΔCT method after values were normalized to those of β-actin. Gene-specific primers are shown in Table 1.

### 2.7. Statistical Analysis

Data were expressed as the mean ± standard deviation (SD), and statistical analysis were performed with SPSS 18.0 software (Chicago, IL, USA). The comparisons among different groups were calculated by one-way analysis of variance (ANOVA) and, further analyzed by Fischer least-significant difference (LSD) test. Values of *p* ≤ 0.05 were considered statistically significant.

## 3. Results

### 3.1. Effects of GEN on HepG2 Cell Viability

To determine cellular toxicity, an MTT assay was performed to ensure that the cell viability remained unchanged after GEN treatment. As shown in Figure 1A, treatment with 0.01–25 μM GEN did not inhibit the growth of HepG2 cells. In addition, cell viability following E2 treatment was measured; 0.1–100 nM E2 had no impact on cell proliferation (Figure 1B). Considering the results and the physiological concentrations of E2 in premenopausal women, 1, 10, 25 μM GEN and 1 nM E2 were selected for further experiments.

### 3.2. Effects of GEN on Lipid Accumulation

As shown in Figure 2, treatment using 25 μM GEN and 1 nM E2 efficiently reduced TG accumulation in HepG2 cells. TG accumulation with 25 μM GEN was approximately 0.6-fold when compared to the control group. In addition, HDL-C content was measured after treatment with GEN; however, HDL-C levels remained unaffected by GEN treatment.

### 3.3. Effects of GEN on Protein Expression Involved in Lipid Metabolism in HepG2 Cells

To determine how GEN improves lipid metabolism, the protein expression of SREBP-1c, FASN, SCD1, CPT1α, PPARα and MTTP, which are essential for lipogenesis, fatty acid β-oxidation and lipid transport, respectively, were analyzed (Figure 3). As shown in Figure 3A, the molecules involved in lipogenesis, such as SREBP-1c, FASN, and SCD1, were all inhibited by the treatment doses (1, 10, 25 μM) of GEN. A total of 25 μM GEN resulted in 0.4-, 0.4-, and 0.6-fold protein expression of SREBP-1c, FASN, and SCD1, respectively, in comparison to the control group. Meanwhile, GEN treatment significantly increased the levels of CPT1α and PPARα protein expression (3.2 and 1.7-fold vs. control group) at 25 μM (Figure 3B), which indicated an improved ability for fatty acid β-oxidation. However, as shown in Figure 3C, no significant changes in the protein expression of MTTP with any dose of GEN treatment were noticed.

### 3.4. Effects of GEN on mRNA Expression Involved in Lipid Metabolism in HepG2 Cells

To verify the genetic regulation induced by GEN, RT-PCR experiments for detecting the mRNA expression of srebp1c, fasn, cpt1α, PPARα, and mttp were performed. As shown in Figure 4, after being treated with different concentrations of GEN, the lipogenic gene expression of srebp1c and fasn were both significantly decreased in HepG2 cells. In addition, pparα gene expression was 1.6-fold that of treatment with GEN when compared to the control group. In addition, the 25 μM GEN treatment noticeably increased cpt1α gene expression (1.4-fold). The expression of mttp, which is mainly responsible for triglyceride protein transfer, was also induced by all doses of GEN treatment in this experiment, in which 25 μM of GEN led to the most significant induction at 1.7-fold relative to the control group.

### 3.5. Effects of GEN on Expression of Key Metabolic Regulators in HepG2 Cells

mTOR, which was recently discovered for its imperative role in lipid regulation, could be an upstream kinase for stimulating the gene expression of srebp1c, fasn, and transcription factors of cpt, pparα [20]. Phosphorylation of Akt was reported to stimulate p-mTOR expression, which means that it is partially involved in the mTOR signaling pathway. Akt phosphorylation could lead to modulation of energy metabolism. To further investigate the mechanisms by which GEN plays a part in regulating lipid metabolism, the phosphorylated activity of Akt and mTOR were examined. As shown in Figure 5A,B, we found that GEN at 10 and 25 μM effectively inhibited the ratio of p-Akt/AKT and p-mTOR/mTOR in HepG2 cells.

### 3.6. The Role of ERβ on Akt/mTOR Signal in GEN-Treated HepG2 Cells

GEN binds to ERs to exert various physiological functions; however, the role of ERβ in hepatic lipid metabolism is uncertain. Since Akt/mTOR plays a vital role in metabolic regulation, the correlation between ERβ and Akt/mTOR expression was analyzed. As shown in Figure 6A, we firstly measured the viability of cells treated with PHTPP, an antagonist of ERβ, and the results showed that 0.001–10 μM PHTPP had no effect on HepG2 cell growth. Considering our MTT results and the effective inhibitive properties on ERβ by PHTPP from another study [21], 1 μM PHTPP was chosen for inducing antagonizing impacts on ERβ expression in later experiments. Further, Figure 6B, on the one hand, demonstrated that the ERβ protein could be expressed in HepG2 cells, but on the other hand ascertained that GEN activated ERβ protein expression, which could be abrogated by pretreatment with PHTPP.

The regulatory functions of ERβ on Akt/mTOR were investigated by treating cells with GEN and/or a pretreatment of PHTPP together. In Figure 6C,D, our results showed that 25 μM GEN inhibited levels of phosphorylation of both Akt and mTOR; however, the inhibitory impacts were reduced in the presence of PHTPP. The results indicated that the activation of ERβ stimulated the regulation of the Akt/mTOR signal.

### 3.7. The Role of ERβ on Hepatic Lipid Metabolism in GEN-Treated HepG2 Cells

To further confirm whether the effects of GEN on lipid metabolism were mediated via ERβ activation, expressions of proteins and genes correlated with lipogenesis, fatty acid oxidation, or lipid transfer were detected using the ERβ antagonist PHTPP. As shown in Figure 7A, protein expressions of SREBP-1c and FASN were downregulated by GEN, whereas the expressions were restored after pretreatment with PHTPP. Moreover, SCD1 protein expression regulated by GEN was reversed by adding PHTPP as well, which is in accordance with the results of SREBP-1c and FASN, which are related to lipogenesis. Meanwhile, the protein expression of PPARα and CPT1α involved in fatty acid β-oxidation were upregulated by GEN; similarly, the stimulatory effects were reduced after adding PHTPP to GEN (Figure 7B). Consistently, as shown in Figure 7C, gene expression confirmed that the protein expression patterns were modulated by GEN or GEN+PHTPP. The lipogenic genes of srebp1c and fasn were repressed by GEN, whereas the inclusion of PHTTP restored inhibitory effects. In addition, the gene expressions of cpt1α and mttp, which are responsible for fatty acid β-oxidation and lipid transfer, respectively, were reduced by cells pre-treated with PHTPP in comparison to GEN alone. These results suggested that ERβ activation was mediated in GEN-regulated lipid synthesis, fatty acid β-oxidation and lipid transfer in HepG2 cells.

## 4. Discussion

Although GEN is recognized as a selective estrogen receptor modulator (SERM) in some studies, the underlying mechanisms of how it modulates hepatic lipid metabolism through ERβ is not yet well investigated. In our current in vitro study, the important findings were that GEN, as a major dietary component found in soy, could regulate lipid accumulation and molecules associated with lipogenesis, fatty acid β-oxidation and lipid transfer via ERβ. In addition, we elucidated GEN-regulated hepatic lipid metabolism by activation of the Akt/mTOR signal, which is also modulated via ERβ.

The liver is one of the most important metabolic organs accountable for lipid metabolism. Postmenopausal women are more prone to developing hepatic lipid disorders. In our previous study, we found that the genes srebp1c and fasn, linked to lipogenesis, were upregulated in the liver of ovariectomized rats, whereas the effects were antagonized by soy isoflavone or GEN treatment [12,22]. In this study, we used an in vitro model of HepG2 cells to investigate the underlying mechanisms of GEN on hepatic lipid metabolism. The doses of GEN we selected were mainly chosen by considering the amount of daily dietary GEN intake [23,24,25], published in vitro studies that also focused on the effects of GEN [26,27,28], as well as our MTT results. The average dietary intake of GEN was about 0.47 μM/kg bw.d for people aged 45+ in China, according to an investigation [23]. In addition to that, quite a number of studies also used 1, 10 or 25 μM of GEN to elucidate its effects on adipocyte differentiation or fatty acid metabolism [26,27,28]. Moreover, our MTT results showed that cell growth was inhibited when treated with a concentration of GEN greater than 25 μM. Further, to prevent impacts by estrogenic substances from culturing medium and FBS, we used phenol-red free DMEM medium and c/d-FBS for at least one time cell passage before compounds were administrated to the cell culture. We also set a group given E2 at a physiological concentration as a control. It is well known that E2 plays an important role in modulating hepatic lipid metabolism, and studies have shown that supplementation with E2 improves lipid metabolism in HepG2 cells [29,30]. TG accumulation was then measured in this study. Under the pathological condition of fatty acid accumulation by treating cells with palmitic acid (PA), GEN reduced TG accumulation [31], exhibiting its effects on lipids reduction. Our results also demonstrated TG levels were inhibited by GEN treatment, similar to treatment with E2. By comparing the effects of GEN to E2, we elucidated that treatment with GEN could inhibit lipid accumulation, similar to the condition of having circulating E2 in vivo. Several other studies that used ovariectomized rats [12] or HepG2 cells without PA treatment [27,28,32,33] proved that GEN was able to regulate lipid metabolism in their respective experimental models, which was consistent with our results.

Ectopic lipid accumulation in hepatocytes is primarily determined by lipogenesis and lipid catabolism. Since GEN might reduce hepatic lipid accumulation, we further investigated whether the effects induced by GEN were correlated with genes or proteins involved in hepatic lipid metabolism. Regarding lipogenesis in the liver, SREBP-1c is a membrane-bound transcription factor that is significantly regulated in de novo lipogenesis. SREBP-1c controls the expression of several genes involved in the fatty acid synthetic pathway, such as FASN and SCD1 [34], which could be partly proven by our results, as these gene or protein expressions were all downregulated by GEN. Notably, SREBP-1c has transcriptional activity when it is processed into a mature form and transported into the nucleus [35]. The size of the mature form of SREBP-1 is 68 kDa, and the size of the immature form of SREBP-1 is 125 kDa [27]; therefore, the protein expression of SREBP-1c was detected at 68 kDa by Western blotting in our current study. In our study, the protein levels of m-SREBP-1c, FASN and SCD1 were decreased by GEN at 25 μM. Likewise, we also proved that GEN significantly reduced the levels of lipogenic genes such as srebp-1c and fasn. Our results demonstrated that GEN inhibits hepatic lipogenesis, which is in line with our previous study as well as other studies [12,27].

Another route of modulating hepatic lipid metabolism is lipid catabolism; therefore, the accumulation levels of triglycerides in liver were also determined by the capacity of hepatic fatty acid oxidation and lipid transfer from liver to blood. Fatty acid oxidation is regulated by some key transcription factors, such as PPARα [34]. PPARα is highly expressed in the liver and promotes fatty acid β-oxidation by stimulating the transcription of the related rate-limiting enzyme CPT1. In the current study, we showed that hepatic protein expressions of PPARα and CPT1α were both increased. Accordingly, the gene levels of pparα and cpt1α were also significantly upregulated by GEN at 25 μM. The mttp gene codes the directives required for the formulation a protein called microsomal triglyceride transfer protein. This protein is a heterodimeric protein that assists in the transfer of neutral lipids between membranes in vitro. Three main functions of MTTP, lipid transfer, apoB binding and membrane association, have been identified. Here we found that the gene expression of mttp was induced by GEN, whereas the MTTP protein remained unchanged. The results of mttp gene expression were in line with another study using colon cancer cells [36]. The inconsistent results from gene and protein expressions might be due to the protein translation process being affected by post-transcriptional regulation. In summary, our findings strongly demonstrated that GEN promotes fatty acid β-oxidation in the liver to reduce triglyceride accumulation.

Akt/mTOR is a classical signaling pathway involved in many cellular and molecular responses, including cell proliferation, apoptosis and migration [13]. It also plays an important role in energy homeostasis. In our study, we found that the ratios of phosphorylation of Akt and mTOR to total AKT and mTOR were reduced by GEN treatment, which was consistent with the regulation patterns of proteins related to lipogenic or fatty acid β-oxidation. A study using diet-induced obese male rats also showed that soy isoflavones suppress the activity of mTORC1 with a reduction in phosphorylation of Akt in liver [37], which is generally in line with our results. mTORC1 is well known to regulate the SREBP transcriptional network. Moreover, one study using inhibitors of Akt and mTOR demonstrated that the PI3K-Akt-mTOR pathway reduced hepatic lipid accumulation by decreasing lipid synthesis, enhancing fatty acid oxidation, and increasing VLDL assembly and secretion in goose hepatocytes [20], which elucidated the importance of Akt/mTOR signaling in hepatic lipid metabolism; the outcomes were in accordance with our results.

The function of ERα in liver metabolism homeostasis is well established; however, the impact of ERβ on liver homeostasis is not yet clear. In our current study, we demonstrated that activation of ERβ by GEN inhibited p-Akt and p-mTOR protein expression, as well as regulated genes or proteins involved in lipid synthesis or fatty acid β-oxidation in the liver. Even though some studies noted that ERβ is minimally expressed in hepatocytes [38], on the contrary, the presence of ERβ in rat hepatocytes has been well described in several studies [12,39,40]. This might be due to the difference in the subunit type of ERβ that has been detected. ERβ2 was described in a study and proven to coexist with ERα in all analyzed tissues, including liver, brain, lung, etc. [41]. Activation of ERβ by selective agonists has anti-obesogenic effects by preventing hepatic lipid accumulation and reducing lipogenic gene expression levels [42]; similar effects to these were also observed by administered GEN-enriched food in OVX rats [12] In contrast, ERβ knockout in intact and ovariectomized mice showed significant increases in adipocyte size and liver TG accumulation compared to wild types [43].

In our current experiments, we used PHTPP to abrogate the increased activation of ERβ by GEN. According to our findings, ERβ was expressed in our in vitro model of HepG2 cells and could be modulated by GEN or PHTPP. This modulation pattern was inconsistent with several other studies that also showed increased ERβ protein levels by GEN [44,45] or decreased ERβ protein levels by PHTPP [46]. The mechanisms of protein expression change by antagonists remains unclear and might be explained due to a complex form of antagonist-ER depending on a ligand binding which shows a tendency for degradation [47,48]. However, compounds with estrogenic effects, including GEN, were shown to increase ERβ expression. The classical traditional Chinese medicine, Liuwei Dihuang, has an estrogen-like effect and has been demonstrated to be able to increase ERβ expression in VSMCs [49]. GEN could activate ERβ transcriptional activity in differentiated cells [44] or increase ERβ expression by reducing its promoter methylation [45]. In addition to this, our important finding was that activated ERβ was able to inhibit the gene or protein expression of lipid synthesis and promote gene or protein expression of fatty acid β-oxidation in parallel with modulation of the Akt/mTOR signal. Even though there is research that reported that ERβ ligand was able to activate Akt to promote remyelination [50], several studies regarding cancer prevention or treatment demonstrated that activation of ERβ repressed the Akt/mTOR signaling pathway [17,18,19]. The underlying mechanisms of how ERβ activates Akt/mTOR signaling were probably associated with up-regulation of the phosphatase and tensin homolog deleted on the chromosome 10 (PTEN) or its concomitant, another phosphatase, inositol-polyphosphate 4-phosphatase type II (INPP4B). PTEN mediates its main function through dephosphorylation of PIP3, resulting in inactivation of Akt. A study by *Guido* et al. [51] proved that ERβ could bind to the *pten* promoter by occupying Sp1, which is concomitant with an increase in RNA Pol II recruitment, leading to increased pten transcriptional activity. PTEN was initially found to be an important tumor suppressor and was also gradually discovered to deeply participate in lipid metabolism. Therefore, these studies provide evidence that ERβ is potentially able to activate the Akt/mTOR signal, which also strongly supports our results. Although whether activation of PTEN at the transcriptional level has important functions in alleviating lipid disorders by ERβ ligands needs further investigation, our current results clearly demonstrated a correlation between ERβ and Akt/mTOR signals in regulating hepatic lipid metabolism in HepG2 cells. Meanwhile, the activation of ERβ and inhibition of Akt/mTOR could be modulated by treatment with GEN.

## 5. Conclusions

Our study provides new evidence that GEN improves hepatic lipid metabolism mainly via ERβ in vitro. Activation of the transcription factor of ERβ regulates hepatic lipid metabolism by inhibiting the Akt/mTOR signal, further suppressing lipogenesis and promoting fatty acid β-oxidation. The Akt/mTORC1 signal is activated by ERβ and might be a crucial mechanism of GEN to alleviate dysfunctions of hepatic lipid metabolism caused by estrogen deficiency in postmenopausal women. Previous in vivo studies suggested positive effects of GEN or ERβ agonist on modulating hepatic lipid metabolism. However, so far there is no in vivo research that directly demonstrates that GEN regulates ERβ in liver lipid accumulation, indicating more extensive research on the in vivo effects of GEN as well as ERβ needs to be done. Nevertheless, based on the positive effects of GEN in modulating lipid metabolism of previous in vivo studies conducted in OVX rats and the possible underlying mechanisms of activating ERβin the current in vitro study, we proposed a better understanding of utilizing GEN, as a SERMβ that could be found in various plant-derived foods, to prevent disease related to hepatic lipid disorder. Moreover, it is promising to discover novel natural food-derived SERMβ substances targeting ERβ for preventing and treating hepatic steatosis, especially for postmenopausal women.

## Figures and Tables

**Figure 1 nutrients-13-04015-f001:**
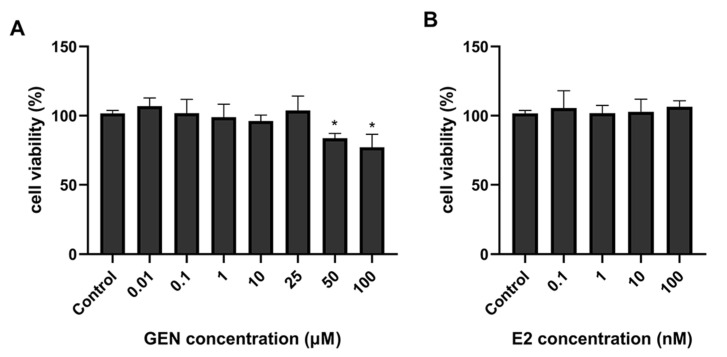
Effects of genistein (GEN) and 17β-estradiol (E2) on HepG2 cell viability. (**A**) HepG2 cells were treated with different concentrations of GEN (0, 0.01, 0.1, 1, 10, 25, 50, 100 μM) for 24 h. (**B**) HepG2 cells were treated with different concentrations of E2 (0, 0.1, 1, 10, 100 nM) for 24 h. Data are presented as means ± SD and analyzed with one-way ANOVA (n = 3). * marks significant differences compared to control group (*p* ≤ 0.05).

**Figure 2 nutrients-13-04015-f002:**
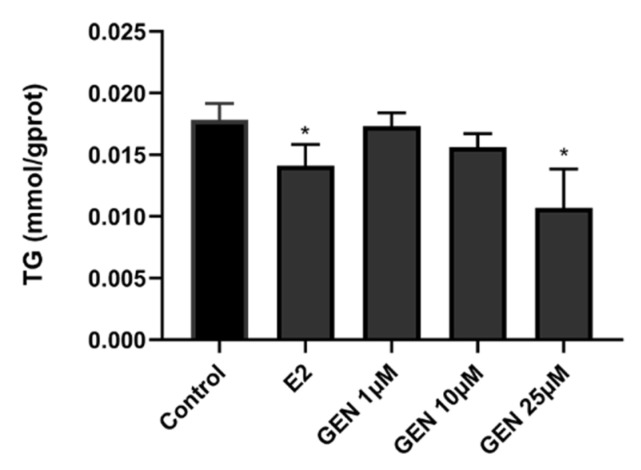
Effects of GEN on triglycerides (TG) in HepG2 cells. HepG2 cells were treated with GEN (0, 1, 10, and 25 μM) and 1 nM E2 for 24 h. Data are presented as means ± SD and analyzed with one-way ANOVA (n = 3). * vs. control group (*p* ≤ 0.05).

**Figure 3 nutrients-13-04015-f003:**
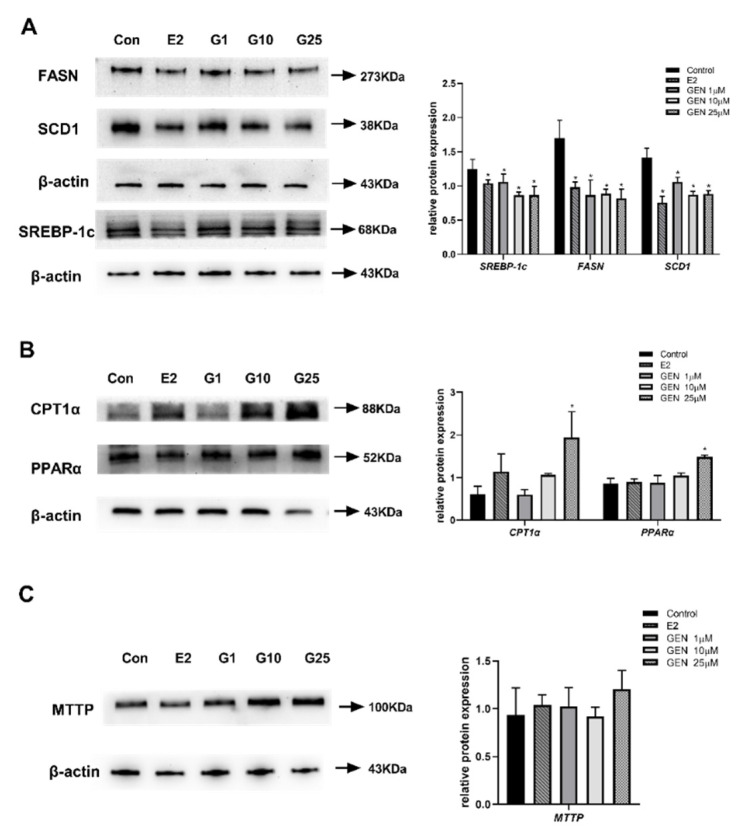
Effects of GEN and E2 on the expression of lipid metabolism-related proteins in HepG2 cells. HepG2 cells were treated with GEN (0, 1, 10, and 25 μM) and 1 nM E2 for 24 h. (**A**) Protein levels involved in de novo lipogenesis and density analysis. (**B**) Protein levels involved in fatty acid β-oxidation and density analysis. (**C**) Protein levels of lipid transfer and density analysis. Western blot bands represent detection of the protein from three independent tests. The relative intensities are expressed in the bar chart. Data are presented as means ± SD and analyzed with one-way ANOVA. * marks significant differences compared to control group (*p* ≤ 0.05).

**Figure 4 nutrients-13-04015-f004:**
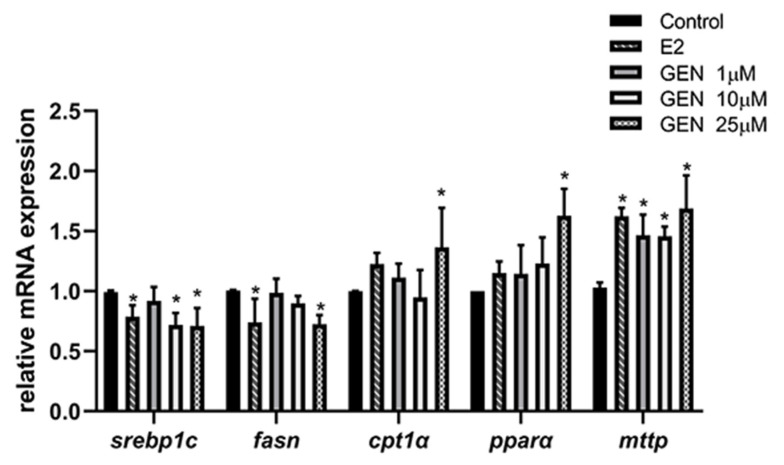
Effects of GEN and E2 on the mRNA expression of lipid metabolism-related genes in HepG2 cells. HepG2 cells were treated with GEN (0, 1, 10, and 25 μM) and 1 nM E2 for 24 h. Expression of gene levels responsible for de novo lipogenesis, fatty acid β-oxidation and lipid transfer were conducted by RT-PCR. Data are presented as means ± SD and analyzed with one-way ANOVA. * marks significant differences compared to control group (*p* ≤ 0.05).

**Figure 5 nutrients-13-04015-f005:**
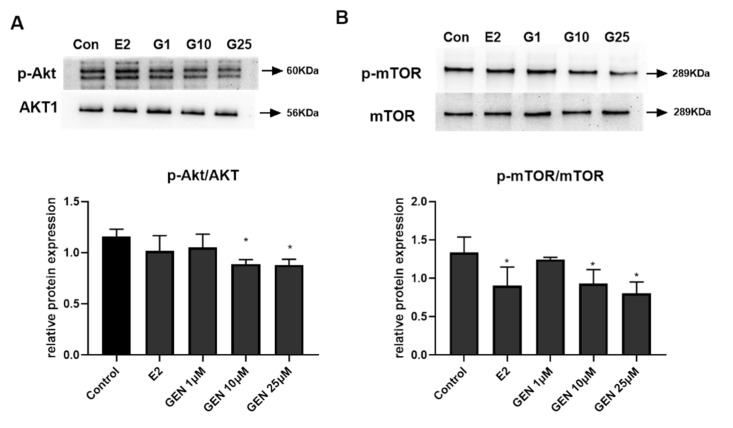
Effects of GEN and E2 on the protein expression of key metabolic regulators in HepG2 cells. HepG2 cells were treated with GEN (0, 1, 10, and 25 μM) and 1 nM E2 for 24 h. (**A**) Protein expression of protein kinase B (AKT)1, phosphorylation of Akt, expression of (**B**) mechanistic target of rapamycin (mTOR), and phosphorylation of mTOR were quantified by densitometry, and the relative intensities are expressed in the bar chart. Western blot bands represent the detection of the protein from three independent tests. The relative intensities are expressed in the bar chart. Data are presented as means ± SD and analyzed with one-way ANOVA. * marks significant differences compared to control group (*p* ≤ 0.05).

**Figure 6 nutrients-13-04015-f006:**
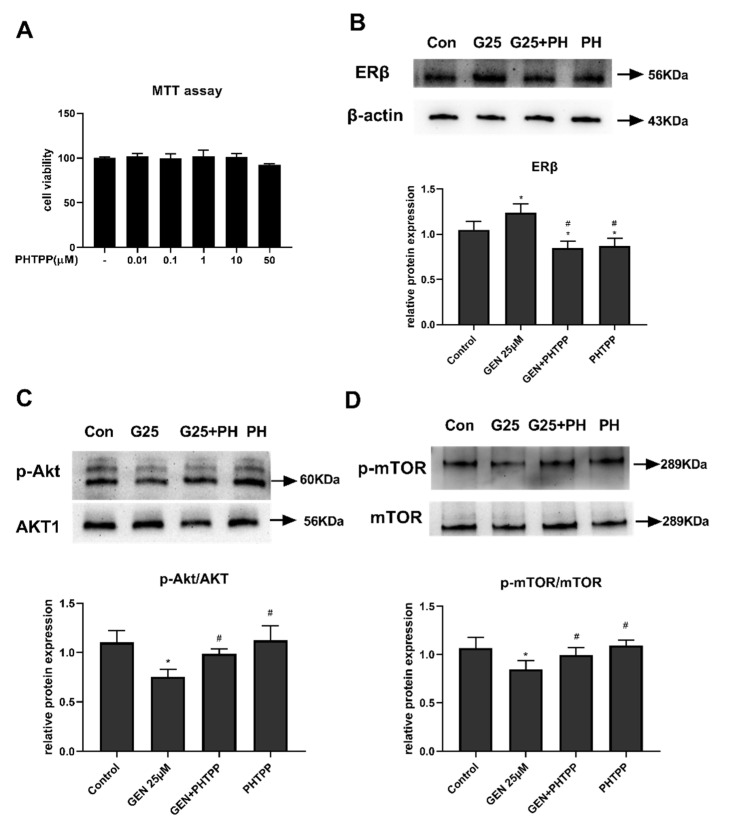
Effects of PHTPP on GEN-treated HepG2 cells. HepG2 cells were divided into four groups: control group, GEN group (treated with 25 μM GEN), GEN + PHTPP group (pretreated with 1 μM PHTPP for 2 h prior to a co-treatment of 25 μM GEN for 24 h), and PHTPP alone group. (**A**) MTT assay for PHTPP on cell viability. HepG2 cells were treated with different concentrations of PHTPP (0, 0.01, 0.1, 1, 10, 50 μM). Protein expression of (**B**) ERβ, (**C**) AKT1, phosphorylation of Akt, (**D**) mTOR, and phosphorylation of mTOR were quantified by densitometry, and the relative intensities are expressed in the bar chart. Western blot bands represent detection of the protein from three independent tests. The relative intensities are expressed in the bar chart. Data are presented as means ± SD and analyzed with one-way ANOVA. * vs. control group (*p* ≤ 0.05); # vs. 25 μM GEN alone treatment group (*p* ≤ 0.05).

**Figure 7 nutrients-13-04015-f007:**
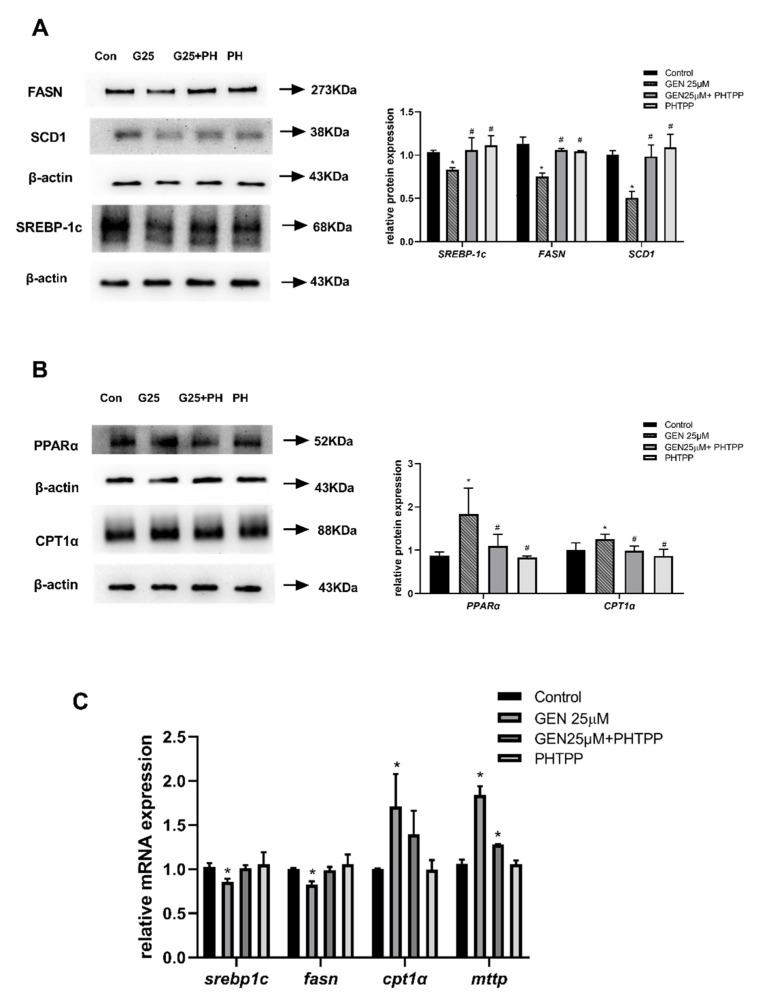
The effect of ERβ on lipid metabolism in GEN-treated HepG2 Cells. HepG2 cells were divided into four groups: control group, GEN group (treated with 25 μM GEN), GEN + PHTPP group (pretreated with 1 μM PHTPP for 2 h prior to a co-treatment of 25 μM GEN for 24 h), and PHTPP alone group. (**A**) Protein expression of de novo lipogenesis and density analysis. (**B**) Protein levels of fatty acid β-oxidation and density analysis. (**C**) Lipid metabolism-related mRNA expression. The relative intensities and relative mRNA expression are expressed in the bar chart. Data are presented as means ± SD and analyzed with one-way ANOVA (n = 3). * vs. control group (*p* ≤ 0.05); # vs. 25 μM GEN alone treatment group (*p* ≤ 0.05).

**Table 1 nutrients-13-04015-t001:** Primer sequences.

Name	Primer Sequence (5′ to 3′)
Srebp1c	forward: GCAACACAGCAACCAGAA
	reserve: GAAAGGTGAGCCAGCATC
fasn	forward: GCCCAAGGGAAGCACATT
	reserve: CGAAGCCACCCAGACCAC
pparα	forward: TAGGGACAGACTGACACC
	reserve: CATAACAAAAGATACGGG
cpt1α	forward: CTACTTCCAGACTTGCCC
	reserve: ACACCATTTCCATTCCAC
mttp	forward: GGAAATGGTCGCTCACAA
	reserve: TGCCAGAACCCGAGTAGAGA
β-actin	forward: TTGCGTTACACCCTTTCT
	reserve: ACCTTCACCGTTCCAGTT

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
