# Peer review of "Genistein Regulates Lipid Metabolism via Estrogen Receptor β and Its Downstream Signal Akt/mTOR in HepG2 Cells"

_nutrients, 2021, doi:10.3390/nu13114015_

Round 1
Reviewer 1 Report
The manuscript is original and provides relevant scientific information. The experimental design is well done. However, some formatting mistakes should be fixed:
In vitro and in vivo should be written in italics.
Authors should take into account that the figures have bad resolution.
p≤0.05 should be written in italics.
Why have the authors selected these doses of Genistein?
Have the authors considered whether the doses selected are physiological?
When gesnistein is ingested in the diet, the digestion process can produce losses or structural changes in the compounds. Are these losses or structural changes taken into account?
Reviewer 2 Report
Qin et al., demonstrated the effect of genistein in regulation of lipid metabolism in HepG2 cells. From the results, the author concluded that genistein regulates lipid metabolism via ERβ and its downstream signal Akt/mTOR. However, the reviewer thinks that there are several major limitation of the study.
Major points
- The author made suggestion under the very restricted condition. What is the meaning of decreasing lipid metabolism under the normal physiological condition? It would be better to give evidence under the pathological condition to verify the effect of genistein. For example, can genistein reduce lipid accumulation under the lipid accumulative conditions? These experiments can be easily done under the fatty acid treatment condition. Furthermore, the author only showed in vitro effect of genistein. Could genistein also modulate ERβ in liver lipid accumulation? These should be addressed to make the clear conclusion of the study.
2 The connection between ERβ and Akt/mTOR signaling is too weak. ERβ mainly locates in the nucleus, and is a nuclear receptor that is activated by translocated estrogen from the circulation. It mainly increases gene transcription, and thus, influences various cellular processes. The author suggested that ERβ increases Akt/mTOR signaling pathway, and gave references in the introduction part. One of the reference article has been withdrawn (Zheng et al.,), which means the article has not been published yet. And another evidence by Yang et al., also does not show strong evidences between ERβ and Akt/mTOR signaling. There must be strong evidences between ERβ and Akt/mTOR signaling to support author’s conclusion. How nuclear receptor such as ERβ can activate Akt/mTOR signaling? Membrane located ERβ is known to change intracellular signaling. The author should clarify this point by adding other references.
3 The author used PHTPP as an antagonist of ERβ. Antagonist means the inhibitor of protein (or receptor), which basically modulate activity of the protein. However, from the Figure 6, the author showed the expression of ERβ has been changed, which only can be explained by increased gene expression or decreased protein degradation. Furthermore, genistein increased protein expression of ERβ. The author should check why the protein expression of ERβ has been changed under these conditions. The reviewer does not understand why the protein expression has been changed by agonist and antagonist
Minor point
- Several Western blot data are not clear and does not show same tendency with the quantification data. For example, which band is true band for the SREBP1c in the Figure 7A? Other western results are also not clear and vague to see the actual differences.
- I see the duplication of Actin bands in figure 6 and 7. Furthermore, actin, which is used as internal control, does not show consistent expression in every different experiment.
- English should be improved.
